# Is There a Role for Levo-Thyroxine for the Treatment of Arterial Erectile Dysfunction? The Clinical Relevance of the Mean Platelet Volume

**DOI:** 10.3390/jcm9030742

**Published:** 2020-03-10

**Authors:** Rossella Cannarella, Aldo E. Calogero, Antonio Aversa, Rosita A. Condorelli, Sandro La Vignera

**Affiliations:** 1Department of Clinical and Experimental Medicine, University of Catania, 95123 Catania, Italy; acaloger@unict.it (A.E.C.); rosita.condorelli@unict.it (R.A.C.); sandrolavignera@unict.it (S.L.V.); 2Department of Experimental and Clinical Medicine, “Magna Graecia” University, 88100 Catanzaro, Italy; aversa@unicz.it

**Keywords:** levo-thyroxine, subclinical hypothyroidism, erectile dysfunction, peak systolic velocity, mean platelet value, mean platelet volume (MPV)

## Abstract

Background: Arterial erectile dysfunction (ED) is an early sign of vascular damage. Rare evidence has been published so far as to whether subclinical hypothyroidism (SCH) affects arterial erectile function. Therefore, the objective of this study was to fill this gap. Methods: Patients with arterial ED and SCH were consecutive enrolled and randomly divided into Group A (*n* = 20) and Group B (*n* = 20). Group A was treated with levo-thyroxine (LT4) at the dose of 1 µg/kg/day for six months, whereas patients of the group B did not receive any treatment. Thyroid stimulating hormone (TSH), free-thyroxine (FT4), peak systolic velocity (PSV), International Index of Erectile Function 5-item version (IIEF-5) score, mean platelet volume (MPV), and total cholesterol were evaluated at enrollment (T0) and after six months (T1). Patients without hypertension, diabetes mellitus, dyslipidemia, not on drugs, and with normal total testosterone (TT) values were included in this study. Results: Group A and B did not differ for age (61.2 ± 4.8 vs. 60.3 ± 5.6 years), body-mass index (28.7 ± 2.5 vs. 28.3 ± 2.6 Kg/m^2^), and serum TT levels (481.2 ± 54.0 vs. 492.1 ± 59.7 ng/dL). At T0, serum TSH levels (6.5 ± 1.2 vs. 6.0 ± 1.0 µIU/mL), FT4 (8.8 ± 0.6 vs. 8.8 ± 0.6 pmol/L), PSV (26.5 ± 1.4 vs. 25.8 ± 2.1 cm/s), IIEF-5 score (8.2 ± 1.7 vs. 9.0 ± 1.7), and total cholesterol (167.8 ± 21.7 vs. 171.6 ± 21.3 mg/dL) did not significantly differ in patients of Group A vs. those of Group B. MPV was significantly higher in Group A than in Group B (12.3 ± 0.3 vs. 11.8 ± 0.7 fL). At T1, Group A showed significantly lower TSH (2.26 ± 0.5 µIU/mL), MPV (9.5 ± 0.3 fL), and total cholesterol (137.8 ± 29.2 mg/dL) and significantly higher FT4 (9.3 ± 0.4 pmol/L), PSV (40.0 ± 2.6 cm/s), and IIEF-5 score (20.2 ± 3.6) compared to pre-treatment values. None of these endpoints showed significant change at T1 compared to T0 in patients of group B. Conclusions: Lt4 therapy is associated with an improvement of the erectile function at the vascular level, a decrease in MPV and total cholesterol. LT4 therapy should be considered in patients with arterial ED and SCH.

## 1. Introduction

Erectile dysfunction (ED) is defined by the permanent or recurrent inability to achieve or maintain a penile erection adequate for sexual intercourse [1]. Its global prevalence has been esteemed up to 76.5%, being higher with increasing age [2].

Erection is a complex mechanism, involving neurologic, hormonal, psychologic, and most of all, vascular factors. ED and cardiovascular disease (CVD) have been regarded as different manifestation of the same disorder and the evidence indicates arterial ED as an early marker of CVD [3,4]. Accordingly, factors leading to atherosclerosis and endothelial dysfunction such as hypertension, dyslipidemia, cigarette smoke, hyperinsulinemia, and diabetes mellitus (DM), result in abnormalities of penile and coronary circulation [5]. Notably, being the penile artery size smaller than that of coronary arteries, the same degree of endothelial dysfunction has a worse impact on blood flow in the penile compared to the coronary district. Thus, the diagnosis of ED precedes coronary artery disease (CAD) and it can be used to prevent CVD development [6].

Thyroid dysfunction has an impact on male sexual disorders. Particularly, most of the studies suggested a relation between ED and overt hypothyroidism [7]. By contrast, little evidence has been published so far on the role of subclinical hypothyroidism (SCH), defined as a mild thyroid function failure diagnosed by increased thyroid-stimulating hormone levels (TSH) and normal free thyroxine (FT4), on ED. The available data suggest an association between these two disorders, but the exact pathophysiology of this link is largely unknown [8]. Interestingly, increased TSH levels have been found to associate with endothelial dysfunction and to decrease nitric oxide (NO) availability in the endothelium [9].

On these premises, the aim of this study was to investigate the relationship between SCH and erectile function, by assessing the effect of levo-thyroxine (LT4) therapy on the penile blood flow, mean platelet volume (MPV), and total cholesterol in patients with SCH and arterial ED.

## 2. Subjects and Methods

### 2.1. Patient Selection

This is a randomized controlled study performed in patients referring to the Division of Andrology and Endocrinology, University of Catania, for ED. Forty Caucasian patients with arterial ED and SCH were consecutively enrolled. They did not have diabetes mellitus (DM), hypertension, dyslipidemia, and did not take any drug. Patients were randomly divided into Group A (*n* = 20) and Group B (*n* = 20). Group A was prescribed LT4 at the dose of 1 µg/kg/die for six months. Patients of the Group B did not receive any treatment. The following endpoints were assessed at enrollment (T0) and six months later (T1): body mass index (BMI), total testosterone (TT) TSH, free-thyroxine (FT4), MPV, total cholesterol, peak systolic velocity (PSV) values, and the International Index of Erectile Function 5-item version (IIEF-5) score.

### 2.2. Hormonal Measurements

Hormone evaluation was performed by electro chemiluminescence (Hitachi-Roche equipment, Cobas 6000, Roche Diagnostics, Indianapolis, IN, USA). Reference values were as follows: TSH0.34-4.2 µIU/mL, FT4 6.8–16 pmol/L, TT 47.8–980 ng/dL, total cholesterol 0–200 mg/dL, MPV7.2–11.1fL.

### 2.3. International Index of Erectile Function 5-Item Version

The IIEF-5 provides a validated self-reported measure of ED. A score greater than 21 excludes ED. For scores ranging between 17–21, 12–16, 8–11, or 5–7, the ED is of low, moderate-low, moderate, or severe entity [10].

### 2.4. Ultrasound Evaluation

Dynamic ultrasound of the penile arteries with pulsed Doppler analysis following intracavernous administration of 20 μg of alprostadil (Caverject; Pfizer, New York, NY, USA) was performed for the diagnosis of arterial ED. Ultrasound examination was undertaken with a GX MegasEsaote (EsaoteSpA, Genoa, Italy) device, equipped with linear, high-resolution, and high-frequency (7.5 to 14 MHz) probes dedicated to the study of soft body areas, with color Doppler for detecting slow flow and a scanning surface of at least 5 cm. Following injection, PSV was measured every 10 min for 20–30 min. A PSV <30 cm/s and a non-temporal peak systolic progression indicated the occurrence of an arterial disease.

### 2.5. Statistical Analysis

Results are shown as mean ± standard deviation (SD). The normality of the variables was evaluated with the Shapiro–Wilks test. Statistical analysis was performed by one-way analysis of variance (ANOVA), followed by the Duncan’s Multiple Range Test, using SPSS 22.0 for Windows (22.0, SPSS Inc., Chicago, IL, USA). A *p* value less than 0.05 was accepted as statistically significant.

### 2.6. Ethical Approval

This study was conducted at the Division of Andrology and Endocrinology of the teaching hospital “G. Rodolico”, University of Catania (Catania, Italy). The protocol was approved by the internal Institutional Review Board (n. 4/2019) and informed written consent was obtained from each participant after full explanation of the purpose and nature of all procedures used. The study has been conducted in accordance with the principles expressed in the Declaration of Helsinki.

## 3. Results

Group A and B did not differ for age, BMI, and serum TT levels in a statistically significant manner (Table 1).

At T0, Group A and B did not show any statistically significant difference for TSH (Figure 1A), FT4 (Figure 1B), PSV (Figure 2A), IIEF-5 score (Figure 2B), and total cholesterol (Figure 3B). MPV resulted significantly higher in Group A compared to Group B (Figure 3A).

At T1, Group A showed significantly lower TSH (Figure 1A), MPV (Figure 3A), and total cholesterol (Figure 3B) levels and significantly higher FT4 (Figure 1B), PSV (Figure 2A), and IIEF-5 score (Figure 2B) compared to the values at T0.

TSH (Figure 1A), FT4 (Figure 1B), PSV (Figure 2A), IIEF-5 score (Figure 2B), and total cholesterol levels (Figure 3B) did not significantly change compared to T0 in Group B. On the contrary, a higher MPV was found at T1 compared to T0 in Group B (Figure 2A).

Finally, Group A had lower TSH (Figure 1A), MPV (Figure 3A), and total cholesterol (Figure 3B) and significantly higher FT4 (Figure 1B), PSV (Figure 2A), and IIEF-5 score (Figure 2B) compared to Group B at T1.

## 4. Discussion

By excluding patients with known causes of endothelial dysfunction such as DM, hypertension, dyslipidemia, this prospective study showed for the first time a positive effect of LT4 administration on penile blood flow and on factors related to vascular damage such as MPV and total cholesterol in eugonadal patients with arterial ED and SCH. Noteworthy, the benefit of LT4 administration on the erectile function is subjectively perceived from these patients, as derivable from the sharp amelioration of the IIEF-5 score.

Previous studies have reported ED in patients with SCH [8]. Accordingly, increased serum TSH levels reduce the availability of NO, which is known to play an important role in erectile function by providing corporal smooth muscle relaxation [11]. Thyroid hormones receptors (THRs) are expressed in the penile artery endothelium and in smooth muscle cells [12]. In addition, LT4 administration has been shown to improve the function of both the endothelium and of the smooth muscle cells [13,14] and the responsiveness to sildenafil and prostaglandin E1 in mouse cavernosum muscle [15]. In line with this evidence, the present study indicates that LT4 administration to patients with SCH may resolve arterial ED by improving the blood penile flow (possibly inducing smooth muscle cell relaxation and ameliorating the endothelial function) in patients without diabetes mellitus, dyslipidemia, hypertension, and hypogonadism.

MPV is a prognostic marker in patients with CAD [16]. Indeed, large platelets show increased aggregation, due to the greater number of α granules, containing pro-thrombotic factors [16]. We have previously shown that a pro-aggregation platelet phenotype relates with ED severity. Accordingly, an increasingly higher percentage of platelets expressing the vitronectin receptor (integrin αvβ3) (VR), which is involved in the early phase of platelet aggregation, is associated with lower PSV values in the penile cavernous artery of patients with ED [17]. In addition, meta-analytic data indicate the occurrence of higher MPV values in patients with ED compared to controls [18].

THRs are expressed in hematopoietic stem cells and thyroid hormones can modulate the production of blood cells, including platelets. Thyroid hormones are able to initiate platelet aggregation by the interaction with the VR and endothelium adhesion [19]. This explains the association between overt hyperthyroidism and the risk of thrombosis. However, SCH has also been associated with hypercoagulability, which is reversed after 6 months of LT4 administration [20]. Accordingly, MPV is higher in patients with SCH compared to euthyroidones [21,22,23,24], thus contributing to increasing the risk of CVD in these patients. This evidence largely confirms the results of the present study, providing an indirect mechanism by which LT4 can influence penile blood flow in patients with arterial ED and SCH.

Lipids and endothelium are the main players in the pathogenesis of atherosclerosis. Lipids promote endothelium dysfunction and lipid-lowering regimens, by decreasing the amount of circulating inflammatory markers, are known to improve endothelial function and plaque burden [25]. Similarly, a meta-analysis has shown a significant increase of the International Index of Erectile Function-5 (IIEF-5) score after treatment with statins [26]. As shown by the American Thyroid Association (ATA) guidelines on treatment of hypothyroidism, LT4 administration is effective in reducing total cholesterol levels and the risk of cardiac events in patients aged 40–70 years with SCH [27]. The present study found a reduction of total cholesterol levels in non dyslipidemic patients with SCH and this result associated with the improvement of penile blood flow. Thus, treatment with LT4 seems effective in lowering total cholesterol levels and, therefore, in ameliorating the endothelial function, in patients with arterial ED.

The sharp improvement of the erectile function after treatment with LT4 in eugonadal patients with arterial ED and SCH opens new roads towards the implications of phosphodiesterase 5 inhibitors (PDE-5i) treatment. PDE-5i are well known to act by the increase of NO availability in the penile endothelium. As previously introduced, indirect evidence points to a role of TSH reduction in the increase of NO availability [11], which may likely improve PDE-5i efficacy. No data is currently available on this topic. Further studies are warranted to assess the possible influence of LT4 administration on PDE-5i efficacy.

In conclusion, LT4 administration clinically improved the erectile function by restoring penile blood flow and decreased MPV and total cholesterol levels in patients with arterial ED and SCH. Euthyroidism prevents endothelial dysfunction with direct and indirect mechanisms in patients without other risk factors leading to vascular damage. LT4 administration should be considered in non-diabetic, dyslipidemic, hypertensive, and hypogonadic patients with ED of arterial nature and SCH.

## Figures and Tables

**Figure 1 jcm-09-00742-f001:**
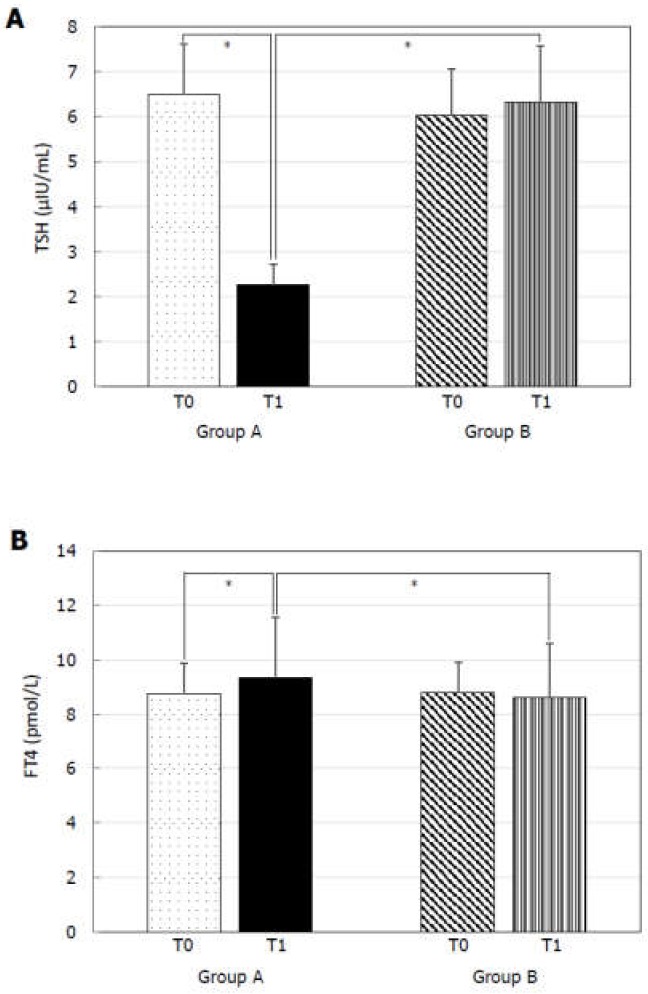
Thyroid-stimulating hormone (TSH) and free-thyroxine (FT4) before (T0) and six months after levo-thyroxine (LT4) or administration or no treatment (T1). Serum TSH levels (**A**) significantly decreased and serum FT4 (**B**) significantly increased after six months of LT4 administration at the dose of 1 µg/kg/day in Group A. * *p* < 0.05.

**Figure 2 jcm-09-00742-f002:**
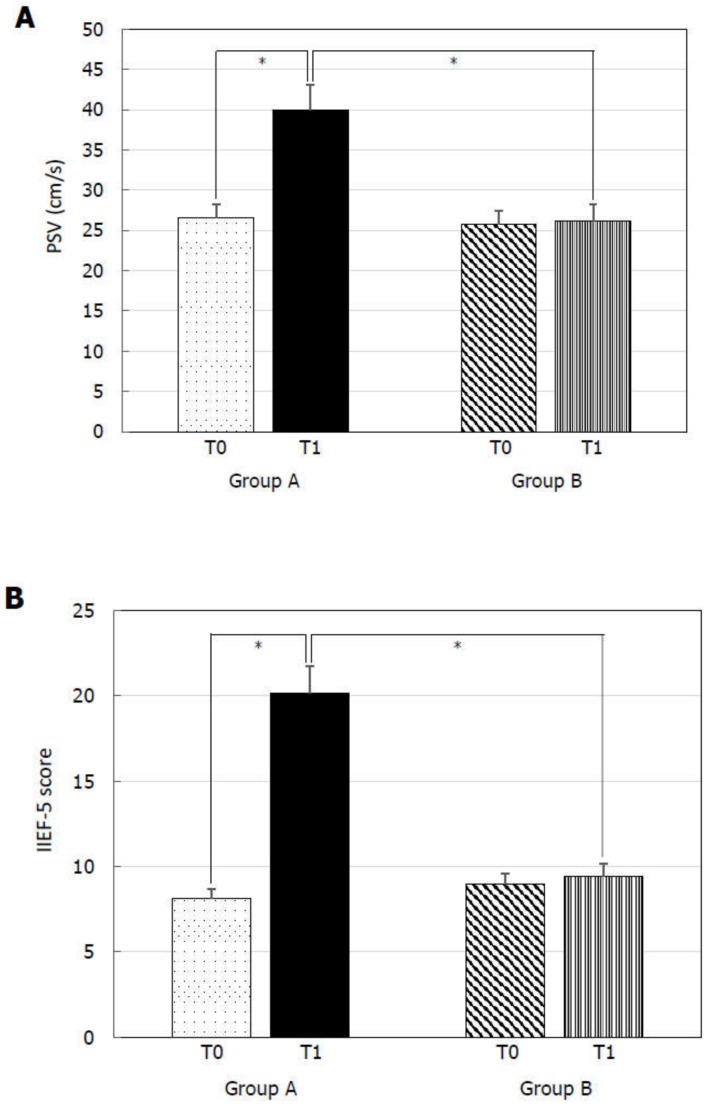
Peak systolic velocity (PSV) of the cavernous artery and International Index of Erectile Function 5-item (IIEF-5) score before (T0) and six months after levo-thyroxine (LT4) or administration or no treatment (T1). PSV (**A**) and IIEF-5 score (**B**) significantly increased in patients of Group A treated with LT4 for six months. * *p* < 0.05.

**Figure 3 jcm-09-00742-f003:**
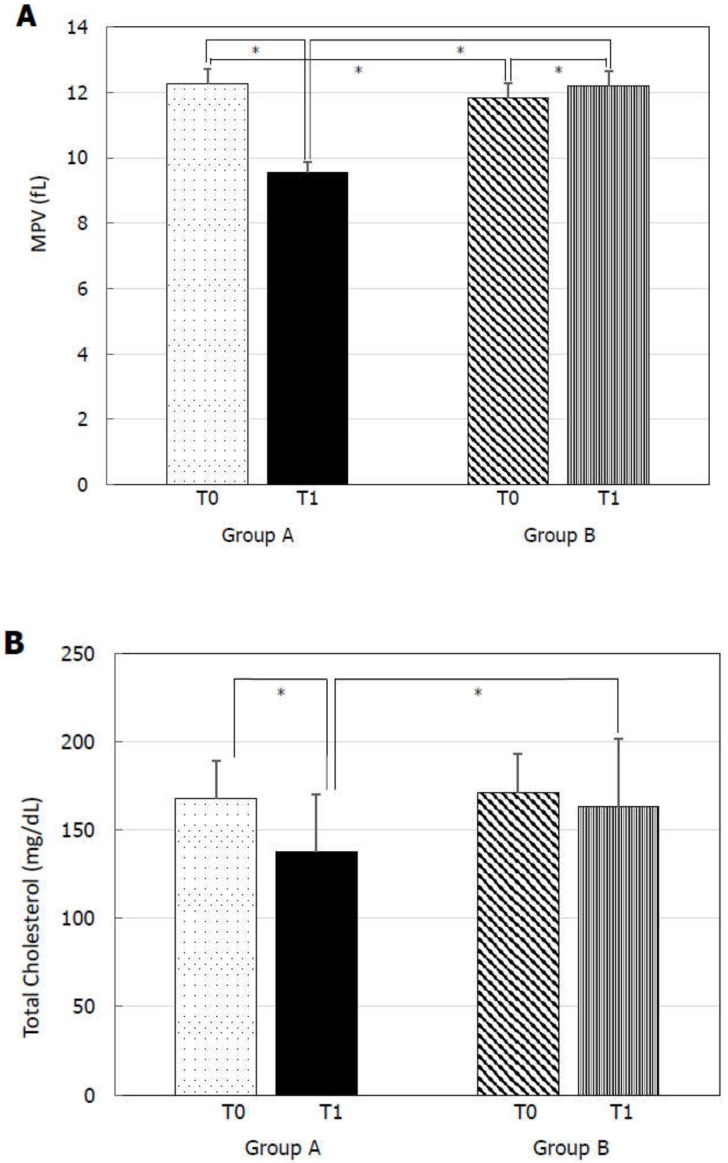
Mean platelet volume (MPV) and total cholesterol at before (T0) and six months after levo-thyroxine (LT4) or administration or no treatment (T1). Serum MPV (**A**) and total cholesterol (**B**) significantly decreased after six months of LT4 administration at the dose of 1 µg/kg/day in Group A. * *p* < 0.05.

**Table 1 jcm-09-00742-t001:** Baseline values of the entire cohort.

Parameters	Group A (*n* = 20)	Group B (*n* = 20)
Age (year)	61.2 ± 8.8	60.3 ± 5.6
BMI (Kg/m^2^)	28.7 ± 2.5	28.3 ± 2.6
TT (ng/dL)	481.2 ± 54.0	492.1 ± 59.7

BMI, body-mass index; TT = total testosterone.

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
