# Peer review of "Is There a Role for Levo-Thyroxine for the Treatment of Arterial Erectile Dysfunction? The Clinical Relevance of the Mean Platelet Volume"

_jcm, 2020, doi:10.3390/jcm9030742_

Round 1
Reviewer 1 Report
I am unclear as to why the authors have not reported the pre and post thyroid level values. Without these, the paper is essentially meaningless; you report that this is a "subclinical" condition, meaning that there are no symptoms to guide the clinician. this means that reporting the labs values is imperative. It seems that you are treating what is only a lab-diagnosed condition, making justification for the intervention questionable.
The figures also need some clarification- I found them very difficult to interpret.
In the discussion, the authors cite reference #11 as support for SCH and ED; however, this paper looked at men with autoimmune thyroid disease rather than SCH. I do not feel this is adequate support for the present study, as autoimmune disease creates a whole different set of stressors than can contribute to ED.
Author Response
Comment 1: I am unclear as to why the authors have not reported the pre and post thyroid level values. Without these, the paper is essentially meaningless; you report that this is a "subclinical" condition, meaning that there are no symptoms to guide the clinician. this means that reporting the labs values is imperative. It seems that you are treating what is only a lab-diagnosed condition, making justification for the intervention questionable.
Answer to comment 1: Thank you for this comment. FT3 levels where within the normal range (as well as the FT4 values, as the Figure 1B shows) both prior and before LT4 therapy. “Subclinical” means that while the FT3 and FT4 are in the normal range, there is a rise in TSH serum levels. The decision to treat SCH is not meaningless but evidence-based. A great amount of evidence (largely discussed the 2014 ATA guidelines on the management of hypothyroidism) points to a possible association between SCH and cardiovascular risk in elderly patients (doi: 10.1089/thy.2014.0028).
Comment 2: The figures also need some clarification- I found them very difficult to interpret.
Answer to comment 2: We have extensively worked on data presentation prior to submit this paper and we realized that this was the more effective way to present them. So, please suggest, if you believe it is necessary, specific comments about how to improve figures. Thank you.
Comment 3: In the discussion, the authors cite reference #11 as support for SCH and ED; however, this paper looked at men with autoimmune thyroid disease rather than SCH. I do not feel this is adequate support for the present study, as autoimmune disease creates a whole different set of stressors than can contribute to ED.
Answer to comment 3: We have removed reference n. 11 (Krysiak R, Szkróbka W, OkopieÅ„ B 2017 The effect of l-thyroxine treatment on sexual function and depressive symptoms in men with autoimmune hypothyroidism. Pharmacol Rep 69(3):432-437)
Reviewer 2 Report
This paper is well thought out and executed. The work is based on a clear hypothesis and the data in general support the notion that LT4 treatment can improve ED in these patients. I only have a few questions and suggestions:
1) Regarding the conclusion in the abstract, the authors state that 'LT4 therapy improved the erectile function by increasing the PSV'. This is an overstatement since the data shows a correlation ie LT4 treated patients have increased PSV, but does not imply a causal relationship. I suggest the authors rephrase that sentence
2) The data are not presented in a very compelling fashion but only as block diagrams at time 0 and after 6 months. How did the time dependence look, fx. what was the results at 3 months?
3) Did the patients use any ED enhancing agents such as PDE-5 inhibitors during the study ? The the patients experience any changes in responde to PDE-5i after LT4 treatment?
4) Did the subjects have any side effect of the treatment?
Author Response
Comment 1: Regarding the conclusion in the abstract, the authors state that 'LT4 therapy improved the erectile function by increasing the PSV'. This is an overstatement since the data shows a correlation ie LT4 treated patients have increased PSV, but does not imply a causal relationship. I suggest the authors rephrase that sentence.
Answer to comment 1: Done as requested. Thank you.
Comment 2: The data are not presented in a very compelling fashion but only as block diagrams at time 0 and after 6 months. How did the time dependence look, fx. what was the results at 3 months?
Answer to comment 2: Unfortunately, the study design provided for the collection of two samples, before and after 6 month the beginning of LT4 therapy. Therefore, actually we cannot provide the data at 3 months.
Comment 3: Did the patients use any ED enhancing agents such as PDE-5 inhibitors during the study? The patients experience any changes in response to PDE-5i after LT4 treatment?
Answer to comment 3: As reported in the Material and Methods section, the use of drugs (including PDE-5 inhibitors) before the inclusion and during the study was an exclusion criterion. Therefore, we couldn’t assess whether the efficacy of PDE5-inhibitors can be influenced by LT4 treatment. We wish to specifically assess this outcome in a further study.
Comment 4: Did the subjects have any side effect of the treatment?
Answer to comment4: No, they did not.